# Ribose-cysteine protects against the development of atherosclerosis in apoE-deficient mice

Tanjina Kader[1¤], Carolyn M. Porteous[1], Gregory T. Jones[2], Nina Dickerhof[3], Vinod K. Narayana[4], Dedreia Tull[4], Sreya Taraknath[1], Sally P. A. McCormick[1]*

**1** Department of Biochemistry, School of Biomedical Sciences, University of Otago, Dunedin, New Zealand, **2** Department of Surgical Sciences, Dunedin School of Medicine, University of Otago, Dunedin, New Zealand, **3** Centre for Free Radical Research, Department of Pathology and Biomedical Science, University of Otago Christchurch, Christchurch, New Zealand, **4** Metabolomics Australia, Bio21 Institute of Molecular Science and Biotechnology, University of Melbourne, Australia

¤ Current address: Cancer Genomics Program, The Sir Peter MacCallum Cancer Centre, Melbourne, Australia
* sally.mccormick@otago.ac.nz

**Data Availability Statement:** The underlying results required to replicate the study can be found in the paper and additional files containing supplementary and original data.

## Abstract

Ribose-cysteine is a synthetic compound designed to increase glutathione (GSH) synthesis. Low levels of GSH and the GSH-dependent enzyme, glutathione peroxidase (GPx), is associated with cardiovascular disease (CVD) in both mice and humans. Here we investigate the effect of ribose-cysteine on GSH, GPx, oxidised lipids and atherosclerosis development in apolipoprotein E-deficient (apoE-/-) mice. Female 12-week old apoE-/- mice (n = 15) were treated with 4–5 mg/day ribose-cysteine in drinking water for 8 weeks or left untreated. Blood and livers were assessed for GSH, GPx activity and 8-isoprostanes. Plasma alanine transferase (ALT) and lipid levels were measured. Aortae were quantified for atherosclerotic lesion area in the aortic sinus and brachiocephalic arch and 8-isoprostanes measured. Ribose-cysteine treatment significantly reduced ALT levels ($p<0.0005$) in the apoE-/- mice. Treatment promoted a significant increase in GSH concentrations in the liver ($p<0.05$) and significantly increased GPx activity in the liver and erythrocytes of apoE-/-mice ($p<0.005$). The level of 8-isoprostanes were significantly reduced in the livers and arteries of apoE-/-mice ($p<0.05$ and $p<0.0005$, respectively). Ribose-cysteine treatment showed a significant decrease in total and low density lipoprotein (LDL) cholesterol ($p<0.05$) with no effect on other plasma lipids with the LDL reduction likely through upregulation of scavenger receptor-B1 (SR-B1). Ribose-cysteine treatment significantly reduced atherosclerotic lesion area by >50% in both the aortic sinus and brachiocephalic branch ($p<0.05$). Ribose-cysteine promotes a significant GSH-based antioxidant effect in multiple tissues as well as an LDL-lowering response. These effects are accompanied by a marked reduction in atherosclerosis suggesting that ribose-cysteine might increase protection against CVD.

**Funding:** This study was supported by funding from Max International, LLC, Salt Lake City, UT, USA who provided the ribose-cysteine and by an Otago School of Medical Sciences Bequest Fund. ST was supported by an Otago Postgraduate Research Scholarship.

**Competing interests:** The funder Max International provided support in the form of a salary for one of the author's conducting the research [TK] and some of the research materials including ribose-cysteine for which they own the patent and commercial license. The funder did not have any role in the study design, data collection and analysis, decision to publish, or preparation of the manuscript. The specific roles of these authors are articulated in the 'author contributions' section. The commercial affiliation with Max International does not alter our adherence to PLOS ONE policies on sharing data and materials. The authors do not have any other competing interests in form of consultancy, patents, products in development, or marketed products, etc.

**Abbreviations:** GSH, Glutathione; GPx, glutathione peroxidase; CVD, Cardiovascular disease; apoE, Apolipoprotein E; ALT, Alanine transferase; LDL, Low density lipoprotein; OxPL, Oxidised phospholipids; NAC, N-acetylcysteine; Ribose-cysteine, D-ribose-L-cysteine; LDLR, Low density lipoprotein receptor; HDL, High density lipoprotein.

## Introduction

Atherosclerosis is the disease process occurring in arteries underpinning the development of cardiovascular disease [1]. Elevated levels of LDL, Lp(a) and remnant lipoproteins have all been established as major risk factors for the development of atherosclerosis in humans [2–4]. Furthermore, the amount of oxidised phospholipids (OxPL) present on these atherogenic lipoproteins associates with vascular disease [5, 6]. Oxidised phospholipids within lipoproteins trapped in the artery promote the activation and infiltration of monocytes that subsequently differentiate into pro-inflammatory macrophages which enable foam cell formation and atherosclerosis development (1).

Oxidised phospholipids are catabolised by glutathione peroxidase (GPx; E.C. 1.11.1.9), an enzyme that catalyses their reduction to lipid alcohols [7]. A low activity of erythrocyte GPx1, the ubiquitous intracellular form of GPx, has been associated with cardiovascular disease in multiple clinical studies [8–10]. Furthermore, a recent meta-analysis of the rs1050450 polymorphism in GPx1 which reduces its activity, is associated with increased CVD risk [11]. An age-related decline in the activity of GPx3, the extracellular form in plasma, has also recently been reported to be associated with cardiovascular disease [12]. The activity of GPx depends on the availability of its cofactor glutathione (GSH), an endogenous tripeptide made from cysteine, glutamine and glycine that provides the reducing equivalents for many redox reactions protecting the body from oxidative stress [13]. Low plasma levels of GSH have been reported in cardiovascular disease patients [14] and in animal models of atherosclerosis [15].

Cysteine-delivery agents such as N-acetylcysteine (NAC) and D-ribose-L-cysteine (ribose-cysteine) can promote the synthesis of GSH [16, 17]. While NAC increases GSH levels at single high doses in humans (around 1.0 g/kg body weight), toxic side effects have been reported [18]. In comparison, ribose-cysteine generates a slower more sustained release of L-cysteine than NAC increasing GSH levels in multiple tissues without any toxicity when administered in a single high dose to mice (2.0 g/kg body weight) [19]. A recent study showed that ribose-cysteine given daily for 8 weeks at a dose of 0.16g/kg/day significantly increased both the liver and plasma GSH levels in a mouse model of hyperlipidaemia without toxicity [20]. Furthermore, this was associated with a significantly increased activity of GPx in the liver and blood and a reduction in the level of oxidised lipids in arteries [20].

The apoE-deficient (apoE -/-) mouse is the most widely used mouse model for atherosclerosis studies with the defective remnant lipoprotein clearance in these animals promoting the spontaneous development of atherosclerotic lesions within 20 weeks [21]. Multiple studies have shown that oxidative stress is a fundamental mechanism underlying the development of atherosclerosis in this model. Plasma and aortic isoprostanes are elevated in these animals [22] and proteomic studies of aortic tissue show the animals lose their ability to mount an antioxidant response with the onset of atherosclerosis [23]. Furthermore, Biwas *et al* [15] showed a reduction in GSH synthesis and GPx1 activity in the aortae of apoE-/- mice before the onset of oxidative stress and atherosclerosis development. The importance of Gpx1 in protecting against atherosclerosis has been established in this model with a knockout of GPx1 accelerating atherosclerosis progression [24].

Here we investigated the effect of ribose-cysteine supplementation on GSH-based antioxidant activity and atherosclerosis development in the apoE-/- mouse. We hypothesised that ribose-cysteine would increase the supply of L-cysteine to the liver, blood and arteries resulting in the promotion of GSH synthesis and increased GPx activity (Fig 1) which may, in turn, reduce atherosclerosis development through a reduction in OxPL content in the arteries.

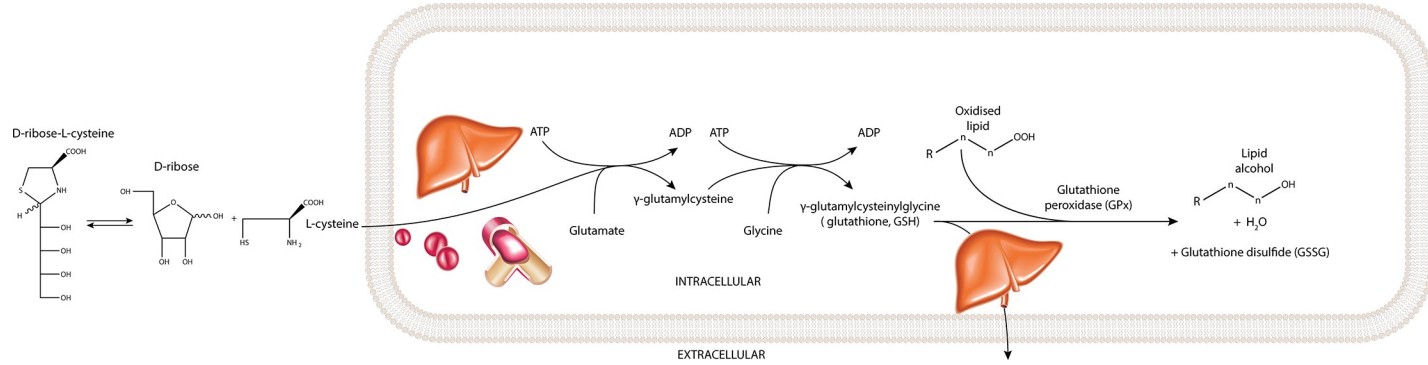

**Fig 1. Proposed mode of action of ribose-cysteine.** Ribose-cysteine releases L-cysteine via hydrolysis in the plasma compartment which is taken up by the liver and utilised to promote GSH synthesis. The increase in GSH promotes an increase in GPx activity to give a reduction in oxidised lipid content. Excess GSH is exported by the liver.

## Materials and methods

### Mice

Female apoE-/- mice on a C57BL/6 background were obtained from the Animal Resource Centre (Murdoch, WA). Ethical approval for this study was granted by the Otago University Animal Ethics Committee. Mice were fed a normal chow-diet (Ruakura 86 Sharpes, Carterton, New Zealand) and housed in a specific pathogen free (SPF) facility on a 12 hour light/dark cycle at 22˚C with free access to water and food. Daily water intake and weekly body weights were recorded. Fifteen 12-week old female apoE-/- mice were treated with 1 mg/mL ribose-cysteine in the drinking water (made fresh every 2–3 days) for 8 weeks receiving an average dose of 4–5 mg/day/mouse (0.16–0.21 g/kg body weight) based on water intake. The stability of ribose-cysteine in water was analysed by HILAC-MS analysis as described in section 2.3. Fifteen 12-week old apoE-/- without ribose-cysteine in their drinking water were used as untreated controls. Ribose-cysteine was prepared by Chemica Inc. (Los Angeles, CA) and provided by Max International, LLC (Salt Lake City, UT).

### Blood and tissue collection

After supplementation, mice were sacrificed by $CO_2$ inhalation and whole blood collected via cardiac puncture into EDTA and plasma and erythrocytes were isolated. Samples were either used fresh or stored at -80˚C with 0.05% butylhydroxytoluene (BHT) until use. Tissues were perfused with heparin 40 U/ml in phosphate buffered saline (PBS) through the left ventricle. Livers from all animals and the aortae from five animals were harvested and either used fresh or frozen in liquid nitrogen and stored at -80˚C with 0.05% BHT until use. The aortae in ten animals of each group were further perfused with 4% (w/v) paraformaldehyde (pH 7.5), carefully dissected out and further fixed in 4% (w/v) paraformaldehyde overnight. The aortae were then washed in PBS and stored in 70% (v/v) ethanol for histological assessment. The aortae from the remaining 5 animals in each group were harvested without fixation and stored at -80˚C with 0.05% BHT until use.

### HILIC-LCMS analysis of ribose-cysteine and related metabolites in plasma

Polar metabolites were extracted from 20 µL of mouse plasma with 180 µL of acetonitrile: methanol (2:2, v/v) containing 5 µM of stable-isotope-labeled ($^{13}$C) internal standards. LCMS analysis of the extracted metabolites was performed as previously described [25]. Briefly,

metabolites (7 μL) were separated on a SeQuant ZIC–pHILIC column (5μm, 150 × 4.6 mm, Merck Millipore) using the Agilent 1200 LC system (Agilent Technologies, Santa Clara, CA) coupled to an Agilent 6545 QTOF mass spectrometer. Ribose-cysteine and its metabolites (D-ribose/any 5-carbon sugar and L-cysteine, see Fig 1) were targeted for analysis. Peak area integration and targeted data matrix was generated on the retention time and molecular masses matching to the authentic standards for each metabolite using MassHunter TOF Quantitative Analysis Software (Agilent Technologies). The stability of ribose-cysteine (20 μM) in water and in plasma *ex vivo* up to 48 hours was tested following the same targeted pHILIC-LC-MS analysis.

## Plasma ALT measurements

Levels of alanine transferase (ALT) were measured in mouse plasma as a marker of hepatotoxicity using the Infinity $^{TM}$ ALT (GPT) Liquid Stable Reagent (Thermo Fisher Scientific, Waltham, MA) according to the manufacturer's instructions.

## GSH measurement

Fresh liver tissue (10–20 mg) was homogenised in 400 μL PBS and following centrifugation, the supernatant was diluted 1:60 with de-ionized water. Fresh plasma was diluted 1:10 with de-ionized water. One hundred μl of diluted homogenate or plasma was mixed with 100 μl of 20 mM NEM to protect thiol groups from oxidation. An isotopically-labelled internal standard of GSH-NEM was added to the alkylated samples, the protein was precipitated by adding ice-cold ethanol (80% v/v) and the protein pellet removed by centrifugation at 12000 g for 5 min. GSH-NEM was measured in the supernatant by stable isotope dilution liquid chromatography tandem mass spectrometry assay (LC-MS/MS) as described before [26]. The GSH content of tissue and plasma was normalized to the amount of homogenized tissue and the protein concentration measured in plasma based on the method by Bradford [27].

## GPx activity

The GPx activity in liver, erythrocytes and plasma was measured using a commercial kit (RS504, Randox Laboratories, Crumlin, UK) with samples prepared according to Kader *et al.* [20]. Plasma was diluted in PBS (1 in 20) before GPx activity measurement.

## Oxidised lipid analysis

Total 8-isoprostanes (free and esterified) were measured in the livers and in the aortic arches pooled from five animals using the EIA kit (Cayman Chemical, Ann Arbor, MI) with samples prepared according to the manufacturer's protocol. Free 8-isoprostanes were measured in plasma using the same kit.

## Lipid analysis

Plasma total cholesterol and triglycerides concentrations were measured using enzymatic reagents from Roche Diagnostics (Mannheim, Germany). High density lipoprotein (HDL) cholesterol concentrations were measured according to Purcell-Huynh *et al.* [28]. The LDL cholesterol concentrations were calculated using the Friedewald equation [29].

## Western blot analysis of LDLR

Liver homogenates (40 μg) were separated by SDS PAGE on 7.5% polyacrylamide gels under reducing conditions and subject to western blot analysis using an anti-low density lipoprotein

receptor (LDLR) antibody (Abcam, Cambridge, ab30532), an anti-HMGCoA reductase antibody (Abcam, ab174830), an anti-SR-B1 antibody (Novus Biologicals, Littleton, CO, NB400-113) and an anti-actin antibody (Sigma, St Louis, MI). Blots were washed and then incubated with a goat anti-rabbit IgG-hrp antibody (Thermo Scientific, Waltham, MA). Membranes were developed using enhanced chemiluminescence (ECL) on the LI-COR Odyssey (LI-COR Biosciences Inc, Lincoln, NE). Protein quantification was performed by Image Studio Lite (LI-COR Biosciences, Inc) with protein normalised against actin.

### Histological assessment of aortae

The aortic arch from the aortic ring to the first intercostal branch was separated from the rest of the paraformaldehyde-fixed aorta. The aortic sinus, from the aortic ring to midway along the ascending aorta, was paraffin embedded for transverse sectioning and the remaining aortic arch embedded separately for longitudinal sectioning. Sequential serial transverse 4-μm sections were made from both the aortic sinus and aortic arch. For the aortic sinus, the transverse section that included at least two of the three leaflet commissures (point of wall attachment) was used for quantification to ensure a similar anatomical site was used for all animals. For the brachiocephalic branch, the longitudinal section with the maximal vessel diameter in each animal was used to avoiding oblique section artefacts and to ensure that a similar anatomical location was used for comparison. Sections were stained with Verhoeff's elastic stain and Curtis' modified van Gieson stain and plaque area quantified, by a blinded assessor, to avoid selection bias as previously described [30].

### Statistical analysis

All statistical analysis except the metabolomics analysis was performed using GraphPad Prism v7 (GraphPad, San Diego, CA). Statistical analysis for the effect of ribose-cysteine on GSH, GPx, oxidised lipids, plasma lipids and lesion area were assessed using an unpaired student $t$-test. All values are expressed as means ± SEM unless specified. A difference with $p < 0.05$ was considered as significant. The metabolomics MS data were processed with Agilent Mass Hunter Software. The data was pre-treated before statistical analysis to account for biological, experimental and instrument variations by performing a natural log transformation and each metabolite was median normalised to the median of each sample. The resulting data (groups) were analysed using a student $t$-test for unpaired data with $p < 0.05$. Results of the $t$-test were controlled for false positives using the Benjamini-Hochberg method. The above statistical test and the generation of box plots of metabolites were performed using the in-house Metabolomics R package.

## Results

### Ribose-cysteine is rapidly metabolised

Ribose-cysteine was shown to be stable up to 48 hours in water and plasma *ex vivo* (S1 Fig). Ribose-cysteine was undetectable in the plasma of treated and control animals suggesting a rapid metabolism in treated animals. The expected metabolites of ribose-cysteine, L-cysteine and D-ribose, were detectable in the plasma with L-cysteine levels unchanged between treated and untreated control mice. While D-Ribose levels appeared to be decreased in treated mice ($p = 0.03$, S2 Fig), this result should be treated with caution since the metabolomic analysis could not distinguish D-ribose from other 5 carbon sugars.

## No evidence of liver toxicity

There was no significant difference between the body or liver weights of ribose-cysteine treated versus untreated control mice (Table 1). Plasma ALT measurements showed the ribose-cysteine treated mice to have significantly lower ALT activity compared to controls (6.46 ± 1.40 *vs* 13.04 ± 0.88 U/mL, $p < 0.0005$, Table 1).

## Ribose-cysteine increases GSH levels in the liver

Liver GSH was significantly increased in ribose-cysteine treated mice compared to controls (6.85 ± 0.43 *vs* 5.48 ± 0.47 μmol/g of tissue, $p < 0.05$, Fig 2A). There was no significant difference in plasma GSH between treated mice and control mice ($p = 0.44$, Fig 2B).

## Ribose-cysteine increases GPx activity in the liver and erythrocytes

The GPx activity in liver tissue was significantly higher in ribose-cysteine treated mice compared to controls (0.98 ± 0.04 *vs* 0.44 ±0.05 U/mg protein, $p < 0.0001$, Fig 2C). Erythrocyte GPx activity was also significantly higher in the treated mice (8.60 ± 1.13 *vs* 3.97 ± 0.56 U/mg protein, $p < 0.001$ respectively, Fig 2D). Plasma GPx activity was not altered in ribose-cysteine treated mice ($p = 0.22$, Fig 2E).

## Ribose-cysteine reduces oxidised lipid levels in the liver and arteries

Analysis of total 8-isoprostanes as a marker of oxidised lipids in the liver showed a significant reduction with ribose-cysteine treatment (118.4 ± 9.7 *vs* 177.8 ± 26.0 pg/mg protein in controls, $p < 0.05$, Fig 2F). Free-8-isoprostanes in plasma were not altered with ribose-cysteine treatment ($p = 0.33$, Fig 2G). Analysis of a pooled sample of five aortic arches also showed a significant reduction in 8-isoprostanes in treated mice (1.26 ± 0.10 *vs* 3.77 ± 0.15 ng/mg protein in control, $p < 0.005$, Fig 2H).

## Ribose-cysteine lowers LDL cholesterol levels

Total plasma and LDL cholesterol concentrations were significantly lower in the ribose-cysteine treated mice compared to control mice (6.15 ± 0.26 *vs* 7.33 ± 0.27 mmol/L, $p < 0.005$ and 5.57 ± 0.23 *vs* 6.68 ± 0.27 mmol/L respectively, $p < 0.005$, Fig 3A and 3B. The concentrations of plasma triglycerides and HDL cholesterol were not significantly different between the two groups (Fig 3C and 3D).

## Ribose-cysteine increases SR-B1 but has no effect on LDLR or HMGCoA reductase protein levels

Western blot analysis of the LDLR and HMGCoA reductase protein showed no difference in protein amounts in the livers of treated versus control mice (S3 Fig). Western blot analysis of the SR-B1 protein showed an increased level in the livers of treated mice (S4 Fig).

**Table 1. Body weight, liver weight and ALT measurements after 8 weeks of ribose-cysteine treatment.**

|  | Control | Ribose-cysteine | p value |
|---|---|---|---|
| Body weight (g) | 22.87 ± 0.31 | 22.80 ± 0.33 | 0.8828 |
| Liver weight (g) | 1.16 ± 0.05 | 1.07 ± 0.05 | 0.2457 |
| ALT (U/mL) | 13.04 ± 0.88 | 6.46 ± 1.40 | **0.0005** |

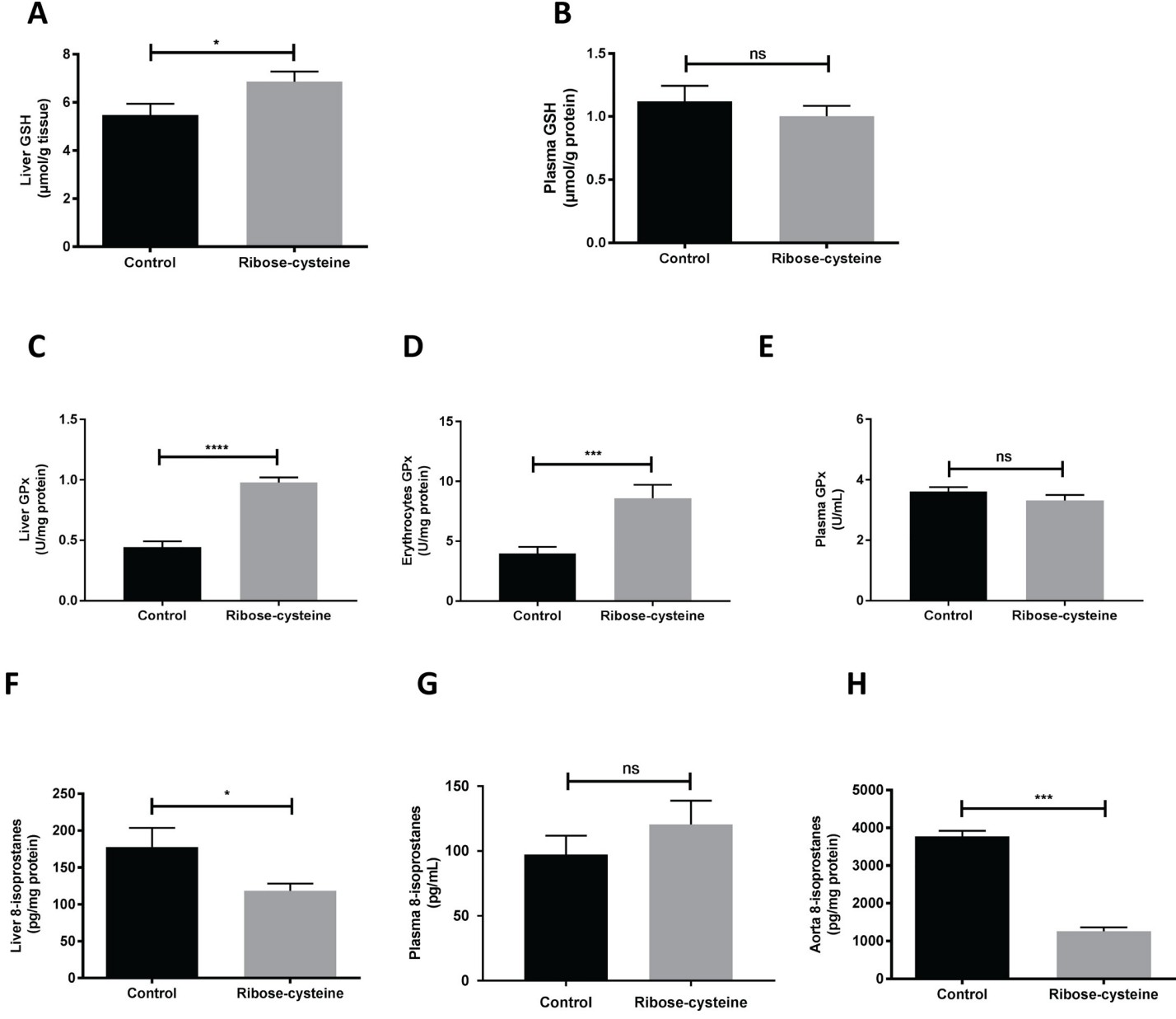

**Fig 2. Ribose-cysteine increases liver GSH and GPx activity and decreases total 8-isoprostanes in liver and aortae of apoE-/- mice.** ApoE-/- (12-week old) were treated with 4–5 mg/day ribose-cysteine in their drinking water for 8 weeks with control apoE-/- mice given normal drinking water (n = 15 per group). The GSH content of tissues and plasma was measured by LC-MS. The GPx activity in liver, erythrocytes and plasma was measured by spectrophotometric assay. Total 8-isoprostanes in liver, aortae and plasma were measured by an 8-isoprostane EIA kit. (A) Liver GSH (n = 15), (B) Plasma GSH (n = 15), (C) Liver GPx activity (n = 15), (D) Erythrocyte GPx activity (n = 15), (E) Plasma GPx activity (n = 15), (F) Total 8-isoprostane in the liver (n = 15), (G) Free 8-isoprostanes in the plasma (n = 15), (H) Total 8-isoprostane in a sample of 5 pooled arteries analysed in triplicate. $^{*}p < 0.05$, $^{***} p < 0.001$, $^{****} p < 0.0001$, ns; not significant. Error bars indicate means ± SEM.

## Ribose-cysteine reduces atherosclerotic lesion area

Histological analysis of arteries showed the ribose-cysteine treated animals to have a significant reduction in the number and size of atherosclerotic lesions in the aortic sinus compared to controls (Fig 4A and 4B). The atherosclerotic lesion area in the brachiocephalic branch of the treated animals was also reduced compared to controls (Fig 4C and 4D). Quantitative analysis showed a significantly reduced lesion area (>50% reduction) in the treated mice compared to

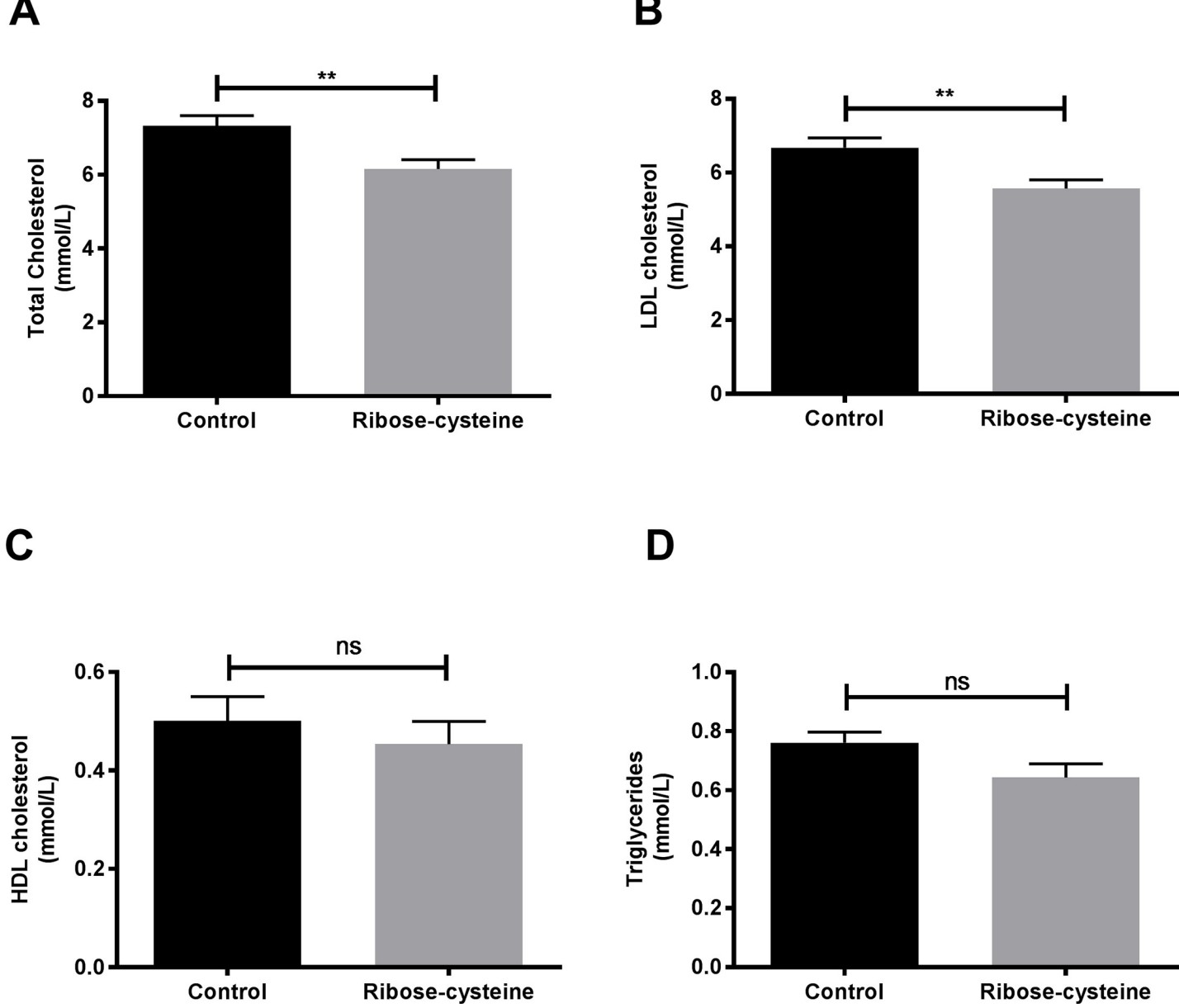

**Fig 3. Ribose-cysteine decreases total plasma and LDL cholesterol in apoE-/-mice.** Total plasma cholesterol, HDL cholesterol and triglyceride levels were measured in ribose-cysteine treated and control apoE-/- mice (n = 15 per group) by enzymatic assay. LDL cholesterol levels were calculated by the Friedewald equation. (A) Total cholesterol, (B) LDL cholesterol, (C) HDL cholesterol, (D) Triglycerides. ** $p < 0.005$. Error bars indicate means ± SEM.

controls both in the aortic sinus ($127.16 \pm 30.64$ *vs* $217.38 \pm 27.74$ $10^3 \mu m^2$, $p < 0.05$, Fig 4E) and brachiocephalic branch ($5.74 \pm 1.75$ *vs* $41.6 \pm 17.1$ $10^3 \mu m^2$, $p < 0.05$, Fig 4F).

## Discussion

Decreased GSH and GPx activity is associated with cardiovascular disease in humans and has been documented in mouse models of atherosclerosis [14, 15]. Here, we show that the cysteine analogue, ribose-cysteine, can enhance GSH and GPx activity in the apoE-/- mouse model of atherosclerosis resulting in protection from disease development. Furthermore, ribose-cysteine showed both hepatoprotective and LDL-lowering effects.

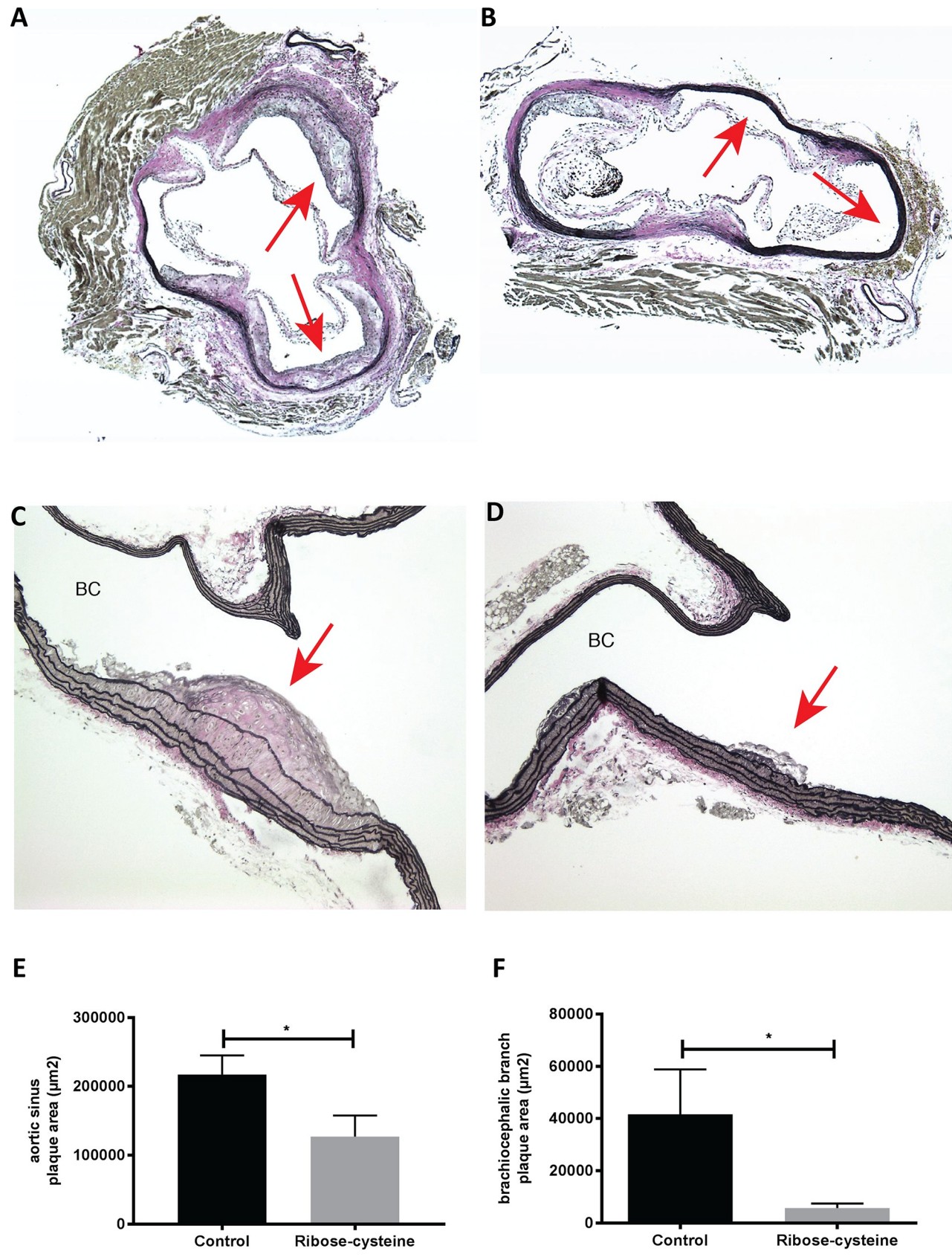

**Fig 4. Ribose-cysteine reduces atherosclerotic lesion area in the apoE-/-mice.** Aortae from apoE-/- mice (20 weeks) were harvested and sectioned for histological staining and measurement of atherosclerotic lesion area. (A, B) Representative staining of the aortic sinus from control and treated mice, respectively. (C, D) Representative staining of the brachiocephalic branch from control and treated mice, respectively. Arrows indicate atherosclerotic lesion area. (E, F) Quantitation of total lesion area in the aortic sinus (n = 9) and brachiocephalic branch (n = 9). *$p < 0.05$. Error bars indicate means ± SEM.

Ribose-cysteine was developed based on its predecessor NAC, a prodrug for delivering L-cysteine, a limiting factor for GSH synthesis in the liver [31]. Elevating GSH protects the liver from damage in situations of acute oxidative stress such as paracetamol overdose [31, 32]. Although still mainly used as an antidote to paracetamol overdose, NAC is currently registered as in use for over 300 trials [33]. Ribose-cysteine is a registered therapeutic goods approved dietary supplement which gives a slower more controlled release of L-cysteine than NAC reducing the potential for toxicity [19]. However, studies on ribose-cysteine are limited and its affect mainly only studied in acute oxidative stress conditions in rodents where it has been shown to increase GSH levels in multiple tissues without toxicity [34, 35]. Only one study has investigated the longer term effect of ribose-cysteine showing that daily supplementation (4 mg/day) in the Lp(a) mouse model of hyperlipidaemia for 8 weeks promotes a significant increase in GSH and GPx in the liver and blood, without toxicity [20].

The current study looked at the effect of 8 weeks supplementation (4–5 mg/day) with ribose-cysteine in the apoE-/- mouse model of atherosclerosis over a time period (20 weeks of age) at which disease develops [21]. ApoE-/- mice of a similar age also display elevated levels of plasma ALT [36] with older mice developing hepatotoxicity and fatty livers [37]. Here we showed that ribose-cysteine is rapidly metabolised (either via hydrolysis in the gut or in circulation) with no accumulation of either L-cysteine or D-ribose in the plasma of treated animals. The released L-cysteine appeared to be readily taken up by the liver to promote GSH synthesis as indicated by the increased liver GSH levels in treated animals. The increased GSH levels may protect against the development of hepatotoxicity as indicated by a significant reduction in plasma ALT levels. This was likely driven by the increase in GPx activity lowering 8-isoprostanes and the potential for oxidative damage of hepatocytes. The increase in liver GSH in the apoE-/- mice, however, unlike the Lp(a) mice [20], did not translate to an increase in plasma GSH levels. The apoE-/- mice displays a more severe lipid phenotype than Lp(a) mice, promoting oxidative stress that likely drives a higher GSH utilisation in the liver leaving less GSH available for export into plasma. Alternatively, an increased requirement for GSH in other tissues under oxidative stress could drive an influx of GSH into other tissues.

ApoE-/- mice show a marked depletion in GSH in the aortic arch before the onset of atherosclerosis due to reduced GSH synthesis [15]. This is associated with a downregulation of GPx1 and increase in lipid peroxidation markers [15]. Mice deficient in GPx1 also have increased levels of aortic isoprostanes along with endothelial dysfunction and structural changes in the artery [38]. GPx1 is expressed in macrophages in the aorta and its expression in peripheral macrophages protects against oxidised LDL-induced foam cell formation [39]. The importance of Gpx1 activity is very apparent in apoE-/-/Gpx1-/-mice which display greatly accelerated atherosclerosis development [24], which is attenuated by treatment with the GPx mimetic, ebselen [40].

In the current study, we show that supplementation with ribose cysteine is associated with a significant reduction in atherosclerotic plaque area in both the aortic sinus and the brachiocephalic arch of apoE-/- mice. This atheroprotective effect is likely attributed to an attenuation of the GSH/GPx depletion previously established in the aortae of these mice as evident by a significant reduction in aortic 8-isoprostanes.

Our study is in keeping with previous studies showing that NAC inhibits atherosclerosis in apoE-/- mice [41, 42]. A more recent study in LDLR-/- mice has shown that NAC effectively inhibits *in vivo* oxidation of LDL [43]. Furthermore, the study showed that NAC was capable of significantly decreasing the level of oxLDL in hyperlipidaemic patients [43]. One effect we established in the ribose-cysteine treated apoE-/- mice, which was not reported for NAC, was that of a significant LDL-lowering effect. This effect had been noted previously in Lp(a) mice treated with ribose-cysteine [20] and seemed to be due to an upregulation in the LDLR. However, in the apoE-/- mice, there were no changes in the protein levels of either LDLR or HMGCoA reductase with ribose-cysteine treatment, suggesting no effect on LDL uptake or cholesterol synthesis via the SREPB2 pathway. These results indicate that an alternative mechanism for LDL-lowering with ribose-cysteine treatment might exist in the apo E-/- model. As a previous study had shown that overexpression of the scavenger receptor, SR-B1, significantly lowered LDL cholesterol levels in chow and fat-fed mice [44], we investigated the protein levels of SR-B1. We saw a significant increase in SR-B1 levels in the ribose-cysteine treated mice which could underpin the reduction in LDL cholesterol levels seen in our study. The LDL-lowering effect of ribose-cysteine may provide further atheroprotection secondary to the GSH-based antioxidant effect.

## Conclusions

In summary, ribose-cysteine is utilised in the circulation to increases GSH levels and GPx activity and appears to provide protection against the development of atherosclerosis. It also displays cholesterol-lowering properties and hepatoprotective effects that could provide further health benefits. Clinical trials of this dietary supplement are needed to evaluate whether these results translate to humans.

## Supporting information

**S1 Fig. Stability of ribose-cysteine (20 μM) in plasma or water up to 48 hours.** The raw response data was normalised to an internal standard ($^{13}C_6$ sorbitol) to account for instrumental variations and the normalised response were plotted over the incubation time. The stability experiments were performed at room temperature.
(DOCX)

**S2 Fig. Ribose-cysteine-related metabolites detected in plasma and compared between the two groups.** The data is presented as mean fold difference in peak areas ±standard deviation after median normalisation. Student t-test with Benjamini-hochberg correction was used for statistical analysis using R (v3.3.1).
(DOCX)

**S3 Fig. Ribose-cysteine has no effect on levels of the low-density lipoprotein receptor (LDLR) and the HMGCoA reductase protein in the liver.** The LDLR (A) and HMGCoA reductase (B) proteins were analysed by western blotting of liver homogenates (40μg) from treated and control mice. Representative blots for 13 control and 14 treated mice are shown.
(DOCX)

**S4 Fig. Ribose-cysteine increases SR-B1 protein expression in the liver.** The SR-B1 protein was analysed by western blotting of liver homogenates (40μg) from treated and control mice. Representative blots for 15 control and 15 treated mice are shown. Fold difference of relative SR-B1 protein, **p<0.01, Kolmogorov-Smirnov test was used for statistical analysis. Error bars indicate means ± SEM.
(DOCX)

**S1 File. Raw data of all main figures.**
(XLSX)

**S2 File. Uncropped western blots.**
(PPTX)

**S3 File. All images of atherosclerotic lesion area in the apoE-/- mice.**
(PPTX)

## Author Contributions

**Conceptualization:** Tanjina Kader, Sally P. A. McCormick.

**Formal analysis:** Tanjina Kader, Gregory T. Jones, Nina Dickerhof, Vinod K. Narayana, Dedreia Tull, Sreya Taraknath, Sally P. A. McCormick.

**Funding acquisition:** Sally P. A. McCormick.

**Investigation:** Tanjina Kader, Carolyn M. Porteous, Gregory T. Jones, Nina Dickerhof, Vinod K. Narayana, Dedreia Tull, Sreya Taraknath, Sally P. A. McCormick.

**Methodology:** Tanjina Kader, Nina Dickerhof, Vinod K. Narayana, Dedreia Tull, Sreya Taraknath, Sally P. A. McCormick.

**Project administration:** Tanjina Kader, Sally P. A. McCormick.

**Resources:** Sally P. A. McCormick.

**Software:** Tanjina Kader, Gregory T. Jones, Nina Dickerhof, Vinod K. Narayana, Sally P. A. McCormick.

**Supervision:** Sally P. A. McCormick.

**Writing – original draft:** Tanjina Kader.

**Writing – review & editing:** Tanjina Kader, Nina Dickerhof, Sally P. A. McCormick.

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
