## [Decision Letter · Decision Letter 0]

17 Oct 2019

PONE-D-19-25760

Ribose-cysteine protects against the development of atherosclerosis in apoE-deficient mice

PLOS ONE

Dear Associate Professor McCormick,

Thank you for submitting your manuscript to PLOS ONE. After careful consideration, we feel that it has merit but does not fully meet PLOS ONE’s publication criteria as it currently stands. Therefore, we invite you to submit a revised version of the manuscript that addresses the points raised during the review process. In particular more experiments are necessary concerning the mechansim of LDL reduction and the verification of the effect on atherosclerosis.

We would appreciate receiving your revised manuscript by Dec 01 2019 11:59PM. To enhance the reproducibility of your results, we recommend that if applicable you deposit your laboratory protocols in protocols.io, where a protocol can be assigned its own identifier (DOI) such that it can be cited independently in the future. For instructions see: http://journals.plos.org/plosone/s/submission-guidelines#loc-laboratory-protocols

We look forward to receiving your revised manuscript.

Kind regards,

Michael Bader

Academic Editor

PLOS ONE

Journal Requirements:

1. The Western blots in Figure S3 are closely cropped. Please provide the original uncropped and unadjusted blots in the figures or supporting information. Please see https://journals.plos.org/plosone/s/figures#loc-blot-and-gel-reporting-requirements

PLOS ONE now requires that authors provide the original uncropped and unadjusted images underlying all blot or gel results reported in a submission’s figures or Supporting Information files. This policy and the journal’s other requirements for blot/gel reporting and figure preparation are described in detail at https://journals.plos.org/plosone/s/figures#loc-blot-and-gel-reporting-requirements and https://journals.plos.org/plosone/s/figures#loc-preparing-figures-from-image-files. When you submit your revised manuscript, please ensure that your figures adhere fully to these guidelines and provide the original underlying images for all blot or gel data reported in your submission. See the following link for instructions on providing the original image data: https://journals.plos.org/plosone/s/figures#loc-original-images-for-blots-and-gels.

2. Thank you for including your funding statement; "This study was supported by funding from Max International, LLC, Salt Lake City, UT, USA who provided the ribose-cysteine and by an Otago School of Medical Sciences Bequest Fund. ST was supported by an Otago Postgraduate Research Scholarship."

We note that you received funding from a commercial source: [Name of Company]

3. Please specify the method of sacrifice in your methods section.

4.

In your Data Availability statement, you have not specified where the minimal data set underlying the results described in your manuscript can be found. PLOS defines a study's minimal data set as the underlying data used to reach the conclusions drawn in the manuscript and any additional data required to replicate the reported study findings in their entirety. All PLOS journals require that the minimal data set be made fully available. For more information about our data policy, please see http://journals.plos.org/plosone/s/data-availability.

Reviewers' comments:

Reviewer's Responses to Questions

**Comments to the Author**

1. Is the manuscript technically sound, and do the data support the conclusions?

Reviewer #1: Yes

Reviewer #2: No

2. Has the statistical analysis been performed appropriately and rigorously? 

Reviewer #1: Yes

Reviewer #2: Yes

3. Have the authors made all data underlying the findings in their manuscript fully available?

Reviewer #1: Yes

Reviewer #2: No

4. Is the manuscript presented in an intelligible fashion and written in standard English?

Reviewer #1: Yes

Reviewer #2: No

5. Review Comments to the Author

Reviewer #1: This is an interesting manuscript about a cysteine analogue producing benefic effects on atherosclerosis disease. This manuscript is a follow up of previous publication of these authors.

The experiments support the manuscript conclusions

The manuscript is clearly enough to be accessible to non-specialists

Histology is impressive; the figures show an important decrease of atherosclerosis plaque in treated mice

Reviewer #2: The manuscript by Kader et al investigated the effect of ribose-cysteine supplementation on GSH-based antioxidant activity and atherosclerosis development in the apoE-/- mouse. However, the experiments and manuscript are poorly conducted and written. The novelty is limited. They have reported that ribose-cysteine increases glutathione-based antioxidant status and reduces LDL in human lipoprotein(a) mice. In the previous paper, they found ribose-cysteine increases LDLR gene and protein expression. In this manuscript, they thought ribose-cysteine does not affect LDLR. More experiments should be conducted to confirm and discuss the effects of ribose-cysteine on LDLR. If LDLR was not affected by ribose-cysteine, they should conduct more experiments to explain the reduced LDL cholesterol by ribose-cysteine. Furthermore, they did not provide enough evidence to support the conclusion that ribose-cysteine reduces atherosclerotic lesion area. The en face lesions of artery should be determined to confirm the effects of ribose-cysteine on atherosclerosis. In addition, more staining assays should be conducted such as oil red O staining.

6. PLOS authors have the option to publish the peer review history of their article (what does this mean?). If published, this will include your full peer review and any attached files.

Reviewer #1: No

Reviewer #2: No

---

## [Author Response · Author response to Decision Letter 0]

4 Jan 2020

We thank the editors(s) and reviewers for their constructive comments and suggestions. All changes are highlighted in Red in the revised manuscript. 

Journal Requirements

1. The Western blots in Figure S3 are closely cropped. Please provide the original uncropped and unadjusted blots in the figures or supporting information. Please see https://journals.plos.org/plosone/s/figures#loc-blot-and-gel-reporting-requirements

We have followed the PLOS ONE requirements for blot results and have provided the original uncropped and unadjusted blots for the revised Figure S3 (and the new figure S4) according to these guidelines in a new additional supporting file (Additional File 3).

2. Thank you for including your funding statement; "This study was supported by funding from Max International, LLC, Salt Lake City, UT, USA who provided the ribose-cysteine and by an Otago School of Medical Sciences Bequest Fund. ST was supported by an Otago Postgraduate Research Scholarship."

We note that you received funding from a commercial source: [Name of Company]

Here is our competing interest statement prepared in line with PLOS One's policy on competing interests:

"The funder Max International provided support in the form of a salary for one of the author's conducting the research [TK] and some of the research materials including ribose-cysteine for which they own the patent and commercial license. The funder did not have any role in the study design, data collection and analysis, decision to publish, or preparation of the manuscript. The specific roles of these authors are articulated in the ‘author contributions’ section. The commercial affiliation with Max International does not alter our adherence to PLOS ONE policies on sharing data and materials. The authors do not have any other competing interests in form of consultancy, patents, products in development, or marketed products, etc."

3. Please specify the method of sacrifice in your methods section.

The method of sacrifice (CO2 inhalation) has been added to the methods section. 

We have added supporting information (original blots for revised Figures S3 and S4, images to support the lesion analysis plus raw lesion analysis data as well as all of the raw data points for all main figures as Additional Files 2 to 4) to fulfil the minimal data set requirement. 

Here is our Data Availability Statement prepared in line with PLOS One's policy on Data Availability:

" The underlying results required to replicate the study can be found in the paper and additional files containing supplementary and original data."

Review Comments to Author:

Reviewer #2: The manuscript by Kader et al investigated the effect of ribose-cysteine supplementation on GSH-based antioxidant activity and atherosclerosis development in the apoE-/- mouse. However, the experiments and manuscript are poorly conducted and written. The novelty is limited. They have reported that ribose-cysteine increases glutathione-based antioxidant status and reduces LDL in human lipoprotein(a) mice. In the previous paper, they found ribose-cysteine increases LDLR gene and protein expression. In this manuscript, they thought ribose-cysteine does not affect LDLR. More experiments should be conducted to confirm and discuss the effects of ribose-cysteine on LDLR. If LDLR was not affected by ribose-cysteine, they should conduct more experiments to explain the reduced LDL cholesterol by ribose-cysteine. Furthermore, they did not provide enough evidence to support the conclusion that ribose-cysteine reduces atherosclerotic lesion area. The en face lesions of artery should be determined to confirm the effects of ribose-cysteine on atherosclerosis. In addition, more staining assays should be conducted such as oil red O staining.

Regarding more experiments to confirm and discuss the mechanism of LDL reduction, we have performed more experiments to investigate the levels of other appropriate targets. We investigated the protein levels of HMGCoA reductase, a key regulator of cholesterol synthesis (see revised Figure S3) and the protein levels were unchanged. Combined with the results for the LDLR, this confirms that the SREBP2a pathway, a key regulator of both proteins was not affected by ribose-cysteine. We also investigated the protein levels of the scavenger receptor, SR-B1 which is implicated in both LDL and OxLDL removal. We saw an increase in the protein level of SR-B1 (see new Figure S4) which may underlie the reduction in LDL cholesterol levels. 

We have added the revised Figure S3 and new Figure S4 and have added further discussion on the possible mechanism of LDL-lowering (see page 18 of the revised manuscript including the addition of a relevant reference (reference 44).

Regarding the comment that not enough evidence was provided to support the conclusion that ribose-cysteine reduces atherosclerosis lesion area, we point out that this comment is in complete contrast to reviewer 1 who thought the histology had been done well and was impressed with the lesion analysis. Nevertheless, we have provided the images from individual animals for both the aortic sinus and brachiocephalic branch and the raw data for lesion area analysis for both regions in additional data files (Additional Files 2 and 4) to satisfy this critique. We have also detailed in the methods (page 9) how the anatomical location of the sections that were quantified from each mouse were chosen to ensure assessment of a similar anatomical location for an accurate comparison. The reviewer also suggested that we should have used the en face method and also stained our current sections with oil red O. En-face staining requires fresh tissue but the histological assessment we choose to perform requires formalin fixed tissue so unfortunately both assessments cannot be performed. Furthermore, as the lipid is removed by the embedding solvents used in processing the formalin-fixed tissue for sectioning, we cannot retrospectively stain our sections with oil red O. The assessment we did perform using histological stains for elastin and collagen is a well-accepted and published form of atherosclerosis assessment (see papers from Erling Falk's laboratory which has published many papers on atherosclerosis assessment in mice i.e. Shim et al Arterioscler Thromb Vasc Biol (2011) 31:1814-20 and earlier papers by Falks group in the same journal). Histological assessment allows for a more precise comparison between individual animals (and therefore groups of animals) since the anatomical location of the section used for quantification is pinpointed using anatomical landmarks and kept the same. Furthermore, more information can be gained about the composition and extent of the lesions i.e. foam cells and cholesterol crystals can be seen which can't be visualised with en face staining which only shows fat content. En face staining is not as accurate for determining the extent of atherosclerosis in animals in the earlier stages of atherosclerosis as were our animals at 20 weeks. Most atherosclerosis studies performed on apo E -/- mice are performed in animals significantly older than 20 weeks that have developed atherosclerotic lesions throughout the length of aorta. Here we performed a more targeted analysis to the two regions known to initially develop atherosclerosis. We hope that the additional data and additional detail we have added to the methods, along with the justification of why we performed a histological assessment rather than use the en-face method, will satisfy this point.

---

## [Editor Report · Decision Letter 1]

15 Jan 2020

Ribose-cysteine protects against the development of atherosclerosis in apoE-deficient mice

PONE-D-19-25760R1

Dear Dr. McCormick,

We are pleased to inform you that your manuscript has been judged scientifically suitable for publication and will be formally accepted for publication once it complies with all outstanding technical requirements.

With kind regards,

Michael Bader

Academic Editor

PLOS ONE
---

## [Editor Report · Acceptance letter]

6 Feb 2020

PONE-D-19-25760R1 

Ribose-cysteine protects against the development of atherosclerosis in apoE-deficient mice 

Dear Dr. McCormick:

I am pleased to inform you that your manuscript has been deemed suitable for publication in PLOS ONE. Congratulations! Your manuscript is now with our production department. 

With kind regards,

on behalf of

Prof. Michael Bader 

Academic Editor

PLOS ONE